# Social Acceptance of Carbon Capture and Storage (CCS) from Industrial Applications

**Katja Witte**

Wuppertal Institute for Climate, Environment and Energy, Division Future Energy and Industry Systems, Doeppersberg 19, 42103 Wuppertal, Germany; katja.witte@wupperinst.org; Tel.: +49-202-2492-218

**Abstract:** To limit global warming, the use of carbon capture and storage technologies (CCS) is considered to be of major importance. In addition to the technical–economic, ecological and political aspects, the question of social acceptance is a decisive factor for the implementation of such low-carbon technologies. This study is the first literature review addressing the acceptance of industrial CCS (iCCS). In contrast to electricity generation, the technical options for large-scale reduction of $CO_2$ emissions in the energy-intensive industry sector are not sufficient to achieve the targeted GHG neutrality in the industrial sector without the use of CCS. Therefore, it will be crucial to determine which factors influence the acceptance of iCCS and how these findings can be used for policy and industry decision-making processes. The results show that there has been limited research on the acceptance of iCCS. In addition, the study highlights some important differences between the acceptance of iCCS and CCS. Due to the technical diversity of future iCCS applications, future acceptance research must be able to better address the complexity of the research subject.

**Keywords:** carbon capture; acceptance; public perception; industrial applications; literature review; knowledge; awareness; communication

## 1. Introduction

To limit global warming to 1.5 °C, the use of carbon capture and storage technologies (CCS) is considered to be of major importance [1–5]. In international parlance, CCS stands for a mix of technological processes for $CO_2$ capture and storage. These are large-scale processes in which carbon dioxide ($CO_2$) is captured from huge $CO_2$ point sources. The captured $CO_2$ is transported via pipeline, ship, or heavy transport and then either reused or injected underground into a suitable geological formation (onshore or offshore) [6].

The use of $CO_2$ capture processes is feasible both in fossil-fired power plants for electricity generation and in energy-intensive industrial processes (for example, steel or cement plants) and could enable a significant reduction in $CO_2$ emissions in these applications. According to the International Energy Agency [7], fossil-fired power plants accounted for about 42.5% of total global $CO_2$ emissions in 2013. In comparison, the share of $CO_2$ emissions caused by industrial activities was around 25%.

In recent years, the discussion around CCS has increasingly focused on its use in the context of industrial facilities (in the following, the term "industrial CCS" is referred to as iCCS). This is mainly because the technical options for the extensive reduction of $CO_2$ emissions in the area of energy-intensive industries without the use of iCCS are not sufficient to achieve the targeted GHG neutrality in the industrial sector. Ref. [4] However, what exactly distinguishes the term iCCS from the classic CCS application? Fossil fuels are an essential input to the production process of the steel, cement, lime and chemical industries, the so-called energy-intensive industries. These fuels are used in the industries for their chemical and physical properties rather than as a primary energy source for power generation, as is the case with CCS [8]. However, unlike electricity generation, it is not possible to replace fossil fuels with renewable energy sources to reduce emissions. This literature review focuses explicitly on the application of CCS to these industrial processes.

The debate to date on the commercial introduction of CCS in fossil-fired power plants (abbreviated below as CCS) has made it clear that numerous other factors are relevant in addition to purely technical and economic indicators. On the part of policymakers, there is a need for a reliable agreement and strategy on the future role of CCS, taking into account international developments around CCS as well as other technological climate protection paths. This will create planning and legal certainty for industry and society and enable the early development of $CO_2$ infrastructure.

Another essential factor, which is the focus of this publication, is the social perception of iCCS technologies and the possible assessment of their future acceptance. Previous research on CCS acceptance has made it clear that CCS technologies may meet with strong opposition, especially in regions where the applications have been tested or were intended to be deployed on a long-term, permanent basis [9,10]. For example, in Germany and the Netherlands, some projects to explore potential $CO_2$ storage formations were abandoned early, primarily due to massive opposition from local communities [11,12]. Since the early 2000s, the number of scientific publications on the acceptance of CCS has continuously increased (see also Section 3). The perception and acceptance of CCS is strongly dependent on the respective country [13] and due to the low level of knowledge about CCS [14,15], it remains difficult to make valid predictions about how specific local attitudes towards CCS might develop.

This study is the first literature review to address the acceptance of industrial CCS (iCCS). The objectives of this study are fourfold. First, it examines the extent to which iCCS acceptance has already been empirically studied. Second, an analytical framework is proposed to systematically review the existing literature. Third, factors that influence iCCS acceptance are identified and discussed based on the review. Fourth, the results on the acceptance of iCCS are compared with the acceptance of CCS in the context of fossil-fired power plants. The assumption is that the attitude of society towards iCCS differs from the attitude towards CCS along individual process steps and value chains. In this regard, first scientific findings are emerging [16,17]. It is unclear in which direction these attitude differences tend.

This study's results should not only contribute to the scientific discussion and further development of the research field, but also hopefully feed into the ongoing practical iCCS discourse in industry and politics. At the international level, there are already associations of industry players testing different technical use cases for iCCS in the form of pilot projects, for example the European Cement Research Academy (ECRA). In some industrial processes, the capture of $CO_2$ emissions is already practiced today, and currently the first projects are underway worldwide in different sectors, such as chemicals (Illinois Industrial), iron and steel (Abu Dhabi Phase 1), and hydrogen (QUEST) [18]. The results of this literature review should also provide indications of possible communication and empowerment needs on the part of the general public and at the same time enable the more technology-based scientific disciplines to place their developments on iCCS in a broader societal context.

In order to be able to better classify the present analysis, the technological component of the research object should first be explained in more detail. For a better understanding of this, Renn's classification [19] of the three areas of technology and their acceptance parameters is helpful. He distinguishes between (1) products—everyday and leisure technology; (2) technology in working life; and (3) external, large-scale and risky technology. The three technology areas differ in terms of their acceptance testing criteria. In the case of current acceptance research on carbon capture and usage (CCU), for example, the focus is often on the concrete evaluation of an end product, which can often be explained in terms of buying or not buying, manageability, long-term durability or direct physical risks (although the research approach here is also broader, for example [20–22]). In the context of the present analysis, all scientific publications dealing with acceptance research on concrete end products (e.g., mattresses, fuels) of CCU technologies were explicitly excluded. This also appears consistent with [23], who clarify that $CO_2$ utilization is often compared and contrasted with CCS; however, they are two different technology pathways so it is necessary

to address and evaluate these technologies separately. Since the subject of the present analysis is the broader society, technology area 2, which deals with technology in the workplace and thus targets "employees", can also be excluded. Following the exclusion principle, only studies dealing with iCCS as an external, large-scale and risky technology (area 3) were analyzed here. For this technology area, the test criteria of acceptability are, for example, societal interests, rights, responsibilities, and legitimacy issues. The focus of this review is therefore on technology pathways that capture $CO_2$ on a large scale and transport it for further purposes without further differentiating whether and how the $CO_2$ is further used.

This paper is structured as follows. First, Section 2 presents the selection of articles analyzed, the methodological approach and the acceptance factors for CCS already identified in the scientific literature, which are also used here as analysis dimensions. The results of the content analysis are explained in detail in Section 3. In the Discussion (Section 4), we present which of the identified acceptance factors for iCCS can be considered crucial for the further development of iCCS and which scientific implications the results induce. The conclusions in Section 5 illustrate some rough propositions for relevant groups of actors dealing with issues of societal acceptance on iCCS in the future.

## 2. Materials, Methods and Acceptance Factors

In order to assess the state of scientific research in the field of acceptance of industrial CCS, a content analysis of scientific articles was conducted. Only articles published in English between 2012 up to and including the end of 2020 were included. This time period was chosen because, to the best of the author's knowledge, no articles were published before 2012 that approached this topic. Thus, the chosen period of analysis seemed sufficient to generate as complete an overview as possible of the state of the scientific literature on this topic.

### 2.1. Selection of Articles

Articles were identified using two online databases. First, the online database of the publisher Elsevier (sciencedirect.com), a full-text database with an inventory of more than 16 million articles and book chapters [24]. Although documents from other scientific publishers are not included, Elsevier is one of the top 5 publishers in the world with over 2000 journals published [25]. Second, the online database was used through scholar.google.com. Google's search engine presents only scientific literature; that is, books or papers from professional journals [26]. Using these two most popular online databases, it was possible to generate the largest possible proportion of scientific literature on the topic of iCCS acceptance.

Only scientific papers, book and conference contributions that could be generated by keyword searches via the two online databases were included in the analysis. In addition, one master's thesis was evaluated that was identified via the online database scholar.google.com and appeared to be relevant. No other dissertations or master's or bachelor's theses were systematically searched for.

Items were identified from November 2020 to 16 January 2021. The following search terms were used to select the technology:

- carbon capture and storage;
- carbon capture;
- CCS;
- carbon capture and storage industry;
- carbon capture industry;
- CCS industry.

The technical search terms were each combined with the following acceptance-related terms:

- acceptance;
- acceptability;
- perceptions;

- attitudes;
- public opinion.

Using a combination of search terms, between 4099 (maximum at sciencedirect.com) and 16,900 (maximum at scholar.google.com) articles were identified in the two online databases. Only articles that explicitly address the topic of industrial CCS were to be included (see Section 1 for narrowing criteria). For further identification of these articles from the existing material, the so-called PRISMA criteria were followed [27]. Based on this procedure, a complete search strategy for one of the databases used is presented below. The presentation is intended to create the prerequisite for the best possible reproducibility of the search.

The search strategy described here as an example refers to the online database scholar.google.com. As previously described, the initial selection was made according to the search terms presented above. With the search term "carbon capture industry acceptance", approximately 16,900 articles were identified on 16 January 2021 (initial access on 8 November 2020). In advance, the search of the articles was restricted to the years from 2012 to 2020 inclusive in the menu under "select period". Subsequently, the search result was sorted by relevance (an option offered by the online database in the menu). The individual short descriptions of the list of results on the homepage were read (not clicked on) and checked to see if all individual search terms were included in the respective text descriptions. This was an indication that all search terms were actually included in the respective target article. In addition, it was checked whether the keywords appeared in the desired context. If, for example, the term "industry" was linked to "coal industry" and the title also indicated that the article was exclusively about CCS as a low-carbon technology for energy generation, the article was excluded from further analysis. The matches identified in this way were further checked for accuracy of fit by reading the respective abstract or, if this did not appear to be sufficient for assessing accuracy of fit, the conclusions.

All hits identified in this way were then included in the pool for further analysis. During the course of the search, it became apparent that after approximately the fourth to fifth page of results on the homepage, the articles listed no longer appeared relevant for the analysis due to missing keywords in the short text. Additional tools from scholar.google.com were used to further identify relevant articles. The option "cited by" lists all articles in which the original hit was cited. A check of these articles was performed according to the criteria already mentioned. The option "related articles" was also used. Using these options, few additional articles could be identified. In addition, an "alert" was created, which was used to automatically notify the author via email when new articles with the given keywords appeared. This option appeared valuable in generating articles that did not appear until the end of the analysis period. To ensure that all articles published by the end of 2020 were identified, a final search query took place in mid-January 2021. The search query at sciencedirect.com followed the same procedure and selection criteria. Beyond the use of the two online databases, a few articles were identified via the references or sources of the articles already identified and read in the course of the evaluation and included in the analysis pool. Using these procedures, a total of 67 articles were identified and included in the closer analysis.

All 67 articles were then read completely. Of these, 42 articles were excluded. There were two main reasons for articles to be excluded:

- Some articles only hinted at possible acceptance conditions for iCCS in their conclusions. A presentation of these references to acceptance seemed mostly comprehensible, but since they could not be sufficiently derived empirically from the study results, the articles were not considered for further analysis.
- Other articles, as part of their methodological approach, focused only on the use of $CO_2$ (CCU) and did not differentiate by source (industrial capture or capture in the context of electricity generation).

Ultimately, 25 articles met the criteria to be included. It can be assumed that a large part of the relevant literature was identified.

### 2.2. Methodical Approach

A qualitative content analysis of 25 articles was carried out using the MAXQDA software. The software allows qualitative data and text analyses and is internationally established in the field of science. For content analysis, a deductive category system was developed (referred to as "analysis dimensions" in the following). It was derived from the previous state of attitude and acceptance research on CCS. During the coding process, some of the analysis dimensions were adapted and the possibility was left open to inductively generate new dimensions, in accordance with the approach of [28]. The individual dimensions or acceptance factors are discussed in more detail in the following subsection.

### 2.3. Acceptance Factors from the Field of CCS

A wealth of individual studies, results, and initial overview studies are available on the perception, attitude, and acceptance of CCS [29,30]. The first studies on the subject appeared from 2002 [31–33]. In the literature up to 2015, publications on the acceptance of CCS focus mainly on the use of the technology in the context of fossil power generation. Therefore, a considerable number of factors determining the acceptance of CCS have been proposed, many of which are commonly used to explain the acceptance of new technologies. There is not a consensus on the one model best suited to predict CCS or technology acceptance [29], although there are publications that present a technology acceptance framework [34] or provide a model approach for selected factors [20,35,36]. Most studies, as mentioned, examine the determining factors along specific research questions that can be categorized into some thematic groups. These groups of topics mainly include (a) general acceptance analyses "of the general public" in one country or in several countries; (b) analyses of real-life-projects across different groups of actors, including the local society; (c) analyses on communication and participation of CCS; and (d) analyses on specific process steps of CCS, especially storage. In recent years, since 2015, more studies have been added on the topic of CCU [20–23,37–41], which can be assigned to the abovementioned group of topics and perhaps also represent a research unit in their own right (cf. chapter 1). However, these factors have predominantly become established and are repeatedly used as a starting point for new research studies and questions. Additionally, for the analysis of the articles identified here for the topic area of industrial CCS, analysis dimensions were generated on the basis of the acceptance factors just mentioned or the state of science (cf. Table 1, here especially the factors from 1 to 8) (a similar set of influencing factors can also be found in the acceptance research on the energy transition [42]). After the initial review of the articles (relevance check), additional dimensions that seemed useful for analyzing the acceptance of industrial CCS were added (compare factors 9 to 11).

**Table 1.** Analysis dimensions of iCCS acceptance within the framework of the review.

| No | Potential Acceptance Factors | Explanation | Source [1] |
|---|---|---|---|
| 1 | Perceived benefits | What personal/societal benefits are associated with iCCS? (social benefits include environmental benefits) | [13,16,43–45] |
| 2 | Perceived risks | What personal/societal benefits are associated with iCCS (including possible costs)? | [13,16,31,44] |
| 3 | Values/attitudes | Can certain patterns of attitudes be identified that have an influence on the acceptance of iCCS? | [34,44,46] |
| 4 | Regional factors | What contribution do regional factors make to the evaluation of iCCS technology? For example, are citizens' previous experiences with potential iCCS companies or local storage options decisive? | [11,12,47,48] |

**Table 1.** *Cont.*

| No | Potential Acceptance Factors | Explanation | Source [1] |
|----|------------------------------|-------------|------------|
| 5 | Trust | How important is trust in iCCS actors for acceptance? What are the reasons for a lack of trust? | [10,41,49–51] |
| 6 | Knowledge/awareness | How does the level of knowledge about iCCS influence the evaluation of the technology? Are initial perceptions of iCCS also important for acceptance? | [52,53] |
| 7 | Communication/participation | What is the need for participatory instruments/communication concepts for the implementation of iCCS? Which communication strategy do companies pursue for marketing/which actors do they involve? | [54–58] |
| 8 | Socio-demographic factors | Can different socio-demographic factors induce distinguished iCCS perceptions? | [44,47,59,60] |
| 9 | Perceived differences to iCCS in the power plant sector | Are there significant differences between the acceptance of CCS in the power plant sector and for industrial applications? | [16,17,41,61–63] |
| 10 | Evaluation according to process step | How is the use of iCCS evaluated along the value chain stages (from investment to capture/transport to $CO_2$ storage and possible reuse)? How is iCCS assessed in the context of other carbon abatement technologies and pathways? | [14,17,41,64] |
| 11 | Regulatory/political aspects | How can a lack of regulatory frameworks, political support and unresolved/complex approval procedures influence iCCS acceptance? | [14,65–67] |

[1] It should be noted that the sources cited in the table are only a small excerpt of possible sources that have dealt with the topic. A comprehensive presentation of studies that have produced results on the respective dimensions of analysis is not intended here. Moreover, the assignment of sources is not exclusive because the respective studies often explored several categories of analysis. In this respect, relevant sources were also assigned to more than one analysis category.

In the following, the results of the evaluated articles are presented along the acceptance factors described in Table 1. In addition to a presentation of the characteristic features, such as methodology used, year of publication and technology path, the analysis clarifies which influencing factors were assumed and investigated to explain the acceptance of industrial CCS. In Section 4 (Discussion), these results are then reflected on and classified in the context of the entire acceptance research on CCS so that first insights can be gained on whether the acceptance factors on iCCS differ from the previous ones, in which areas they differ, if any, and whether new factors have been added.

## 3. Results

### 3.1. Characteristics of the Analyzed Articles

To place the iCCS publications in the overall context of all publications on the topic of CCS acceptance, it should be mentioned in advance that until circa 2014 the number of scientific publications on the acceptance of CCS increased steadily [29]. Between 2015 and around 2018, the number of publications on the topic of CCS acceptance then remained at a lower level than in the years between 2010 and 2014 [30]. Up to this point, publications on the acceptability of CCS focused on the use of the technology in the context of fossil fuel power generation. Triggered by the Paris Agreement 2015 [2], which highlighted the urgency of limiting global warming to as close to 1.5 °C as possible, as well as a number of other publications [1,3–5], as described in Section 1, the discussion about CCS has continuously broadened and has more often focused on technology pathways that are not directly related to fossil energy production. Since then, there has also been an increasing number of scientific publications dealing with the acceptance of different technology paths of CCS.

The articles analyzed here were published between 2012 and 2020. Table 2 illustrates the year of publication of the articles in combination with the selected technology path.

**Table 2.** Theme clusters of iCCS acceptance in combination with year of publication [13–17,30,41,61–78].

| Technology Path | 2012–2015 | 2016 | 2017 | 2018 | 2019 | 2020 | 2021 [1] |
|---|---|---|---|---|---|---|---|
| iCCS without further specification | | Haug et al. [64], Broecks et al. [63] | Pihkola et al. [69] | Xenias et al. [68], Kashintseva et al. [67], Ilinova et al. [70], Thomas et al. [71], van Os [72] | Tcvetkov et al. [30], Whitmarsh et al. [13], Serdoner [73] | Swennenhuis et al. [65], Boomsma et al. [74] | |
| Evaluation of different technology pathways (variation of source, transport, storage) | De Best-Waldhober et al. [17], Wallquist et al. [16], Dütschke et al. [61] | | | | | Offermann-van Heek et al. [41] | |
| iCCS with focus on $CO_2$-storage | | | | Gough et al. [14] | | | |
| iCCS as low carbon technology for energy-intensive industry (cement, steel) | | | | | Aursland et al. [66] | | Williams et al. [62] |
| Bioenergy with CCS (BECCS) | | | Kojo et al. [75] | | Haikola et al. [76] | Rodriguez et al. [77] | |
| iCCS with reference to hydrogen applications | | | | | Alcalde et al. [78] | | Glanz et al. [15] |
| Total | 2/1 | 2 | 2 | 6 | 6 | 4 | 2 |

[1] These two articles have already been published in mid-January 2021. Due to their relevance, the author decided to include them before completing this article at the end of January. No other articles from 2021 were included in the analysis.

As shown in Table 2, by the end of 2020, most articles on iCCS were published in 2018 and 2019 (n = 6 in each year). A slight majority of the 25 articles (n = 13) use the terminology "industrial CCS" (compare row 1 Table 2), but do not further explain which technological concept of iCCS technologies is involved in the definition or within the operationalizations. This is not surprising, as the technological applications of iCCS are highly complex along the process steps and the different value chains that may be involved.

To address this complexity, four of the studies provided their participants with a selection of different realistic CCS technology pathways to evaluate (compare row 2 Table 2), which at least allowed for a more differentiated view according to different $CO_2$ sources, such as the evaluation of $CO_2$ capture in a chemical plant [41]. Since 2019, there has been an increase in acceptance studies investigating the impact of specific industrial CCS applications, such as from cement or steel plants or for the BECCS sector. These studies are often linked to specific project proposals, for example the ALIGN project (It is expected that further scientific publications on the acceptance of iCCUS will be published in 2021 from research projects that have been and will be funded within the framework of Horizon 2020 of the European Commission, such as the ALIGN-CCUS and STRATEGY CCUS projects) [74], and concentrate on regions with industrial clusters that are significant geologically and in terms of their industrial structure with regard to the development of iCCS and are already being scientifically researched in part (compare lines 4 to 6, Table 2).

The analyzed articles on iCCS acceptance come from a total of 15 different countries, of which European countries represented 13—an overwhelming majority. The following European countries were involved in the preparation of the articles: United Kingdom = 7; The Netherlands and Germany = 4 each; Norway = 3; Finland and Sweden = 2 each; and Austria, Belgium, Lithuania, Portugal, Romania, Spain and Switzerland with one article each. Five of the European articles involved more than one country. As mentioned at the beginning, previous studies on the acceptance of CCS have made clear that protests and

risk perceptions on CCS have formed along exploration plans and projects, especially in Europe—particularly in the Netherlands [12] and Germany [11].

In this respect, if an iCCS strategy is to be pursued on the political level in the long term, these countries seem to have a particular interest in predicting future developments regarding the acceptance of iCCS. For Great Britain, the situation is similar; here, according to [79], 17.2% would "probably not use" or "definitely not use" CCS technologies according to a representative survey. A further three articles come from Russia and another one from the United States of America. According to [30], Russia has a special interest in the use of enhanced oil recovery (EOR) technology, which requires a lot of $CO_2$, and therefore is considering CCS as a future option to develop this technology.

The relevant articles on the acceptance of iCCS were published in a wide range of journals. In total, the 25 articles come from 15 different journals. *The International Journal of Green-house Gas Control* accounts for 8 articles—by far the most. This is followed by the journals *Energy Procedia* and *Journal of Cleaner Production*, with 2 publications each on the topic. One of the analyzed articles is a Master's thesis, which was written at the University of Graz and cannot be assigned to any journal [73].

Different theoretical concepts and approaches were used in the articles included. Twelve of the analyzed articles on iCCS acceptance do not mention any theoretical concepts. The concept of Wüstenhagen [80] to classify three different dimensions of social acceptance is mentioned and applied in two articles. Studies that focus their analysis more on the regional or project level often include actor and communication-related approaches, such as the theory of public engagement in [68], the social licence to operate (SLO) in [14,74], the end-to-end stakeholders involvement approach in [67], the concept of procedural fairness in [62], the concept of media agenda-setting in [75], the stakeholder theory for management in [70] and the cognitive theory of shifting coalitions in [73].

In addition, the articles mention social-psychological concepts that illuminate social behavior even more against the background of cultural aspects and certain values, such as the theory of planned behavior in [30] and, in the context of the Master's thesis, the concept of the Ethical landscape of CCS, the theory of worldviews and the cultural theory to specify belief systems in [73]. Two of the analyzed articles reflect their findings on iCCS acceptance to the whole debate on energy system transformation using the just transition approach [65,78] or the multidimensional research concept as in [15].

A complete table of the analyzed articles with the categories "first author", "year of publication", "method(s) used", "country", "iCCS-related technology", and "important statement in relation to iCCS" is provided in the Appendix A (Table A1: Overview of the analyzed articles).

### 3.2. Key Findings along the Dimensions of Analysis as well as Additional Insights

In the following, the main results of the analyzed articles are presented along the analysis dimensions shown in Table 1.

### 3.2.1. Perceived Benefits

The results of the studies analyzed have identified some benefits that appear to be associated with the use of iCCS and thus may have a positive impact on social acceptance. These benefits include the possibility of creating local and national value through iCCS projects [64].

For example, the municipality of Porsgrunn in Norway considers iCCS important in legitimizing industry in the region and thus sustaining related jobs in the long term [64]. Additionally, ref. [71] sum up that the potential of iCCS can protect and rejuvenate historical employment patterns and this opportunity makes iCCS an attractive option for an area. This is also important to counteract the out-migration of the local population that threatens to occur if established industries go away [64]. Beyond protecting existing jobs, ref. [71] make the argument that providing infrastructure for iCCS can also create additional employment opportunities in the region. Consistent with this, communities hosting CCS

projects would benefit economically from the jobs and revenue that the industry would provide [13].

In addition, regional clusters containing multiple capture projects can benefit from shared $CO_2$ transport and storage infrastructure to maximize value, share investment decisions and operating costs, and thus reduce development costs [78]. Thus, ref. [64] postulate benefits from mergers of larger regional clusters for iCCS (across national borders). For example, in their study, they identified the notional "Skagerrak Cluster" for the countries of Norway, Sweden, and Denmark, which identifies some key geographic features that have good conditions for establishing iCCS technology (similar to the northeast region of Scotland). The advantages come from the possibility of storing the $CO_2$ offshore, with emission sources relatively close to the sea. According to [64], the relevance of looking more closely at the Skagerrak cluster provides valuable input for evaluating acceptance and communication challenges for other iCCS clusters in the Nordic region. These benefits of iCCS overall can be linked to increasing the economic viability of both the technology itself and the region in question, these are benefits that [30,70] also highlight in their study.

However, not only is the preservation or renewal of existing economic structures identified as a benefit of iCCS, but the technologies should also serve to promote and profile municipalities and regions as environmental and technological leaders, ultimately to develop new industrial activities [64]. In this context, there is also talk of a potential image boost for iCCS industries and regions [62]. For example, refs. [75,77] argue the relevance of developing and deploying BECCS, a technology pathway discussed as an advantage for forest-rich countries such as Finland [75] and which holds the potential to establish itself as a "first mover" [77]. Without BECCS it would be a challenge to meet emission targets, but with BECCS Finland could gain advantages by saving and trading emission rights [75] (see also Section 3.2.11).

Regarding the impact of environmental effects (reduction of $CO_2$ emissions, slowing of climate change) and their classification as a benefit for the acceptance of iCCS, there are different results in the analyzed studies. Some study results suggest that attributing the benefits of iCCS to improving the regional and global environmental situation can create an advantage for the perception of acceptance [15,30,70,75]. Similarly, the results of a representative study in Canada, the USA, the UK, the NL, and Norway illustrate that iCCS can help mitigate climate change and support the economy according to the respondents in [13], which could be interpreted as a benefit for the technology. However, the same study also highlighted that framing CCS as dealing with 'waste' (in conjunction with $CO_2$ reuse) seems to be more persuasive in encouraging support than framing it in terms of climate or economic benefits. The authors of [74] critically note that the siting of new or expanded iCCS facilities is more likely to be associated with national and international benefits, for example achieving energy and climate goals and economic revenues (on this also see [70]), and that the apparent benefit to local communities may turn out to be a potential burden, for example through subjectively perceived risks. Such a perceived imbalance between (negative) local impacts and national or global benefits would pose a challenge when it comes to public response to iCCS technologies [74]. Hence, currently there is no consistent evidence from the scientific community as to whether iCCS is perceived as a mitigation option for $CO_2$, and thus as a climate technology, and whether this has a positive or negative effect on the perception of the benefits of the technology. Moreover, such a perception is certainly also dependent on many regional factors.

For completeness, here are the five main benefits of CCS industrial projects according to [70]: (1) reduction of negative impacts on the environment, (2) contribution to socio-economic development of regions and territories, (3) attractive direction for socially responsible investments, (4) support for sustainable development of companies involved in CCS projects, (5) use of $CO_2$ for purposes such as improving oil recovery by oil and gas companies, increasing energy efficiency of industrial companies.

The analysis of perceived benefits gives the impression, as also indicated by [30] and previous studies on the benefits and risks of CCS, that benefit perception may exert a stronger influence on iCCS acceptance than risk perception.

### 3.2.2. Perceived Risks

According to the studies analyzed, the use of iCCS technologies is associated with various societal risks that can have a negative impact on acceptance. These include perceived risks at the local level, for subsequent generations and for ecological and economic systems, but also risks for making political decisions that do not contribute to improving climate protection in the long term. The most frequently mentioned risk perceptions in the studies relate to negative health impacts, especially for people living near $CO_2$ storage and transport infrastructure [62].

The local impacts of iCCS are particularly addressed here [68], and with it the accompanying sense of unfair treatment of those who suffer disadvantages [30,74]. It is believed that iCCS could become locally entrenched as a "risky technology" in the perception of local and regional populations [15], especially if $CO_2$ storage occurs on land [77]. Hazards are expected from possible $CO_2$ leakage and seismic risks [15,75,77]. The perception would not improve even if already existing infrastructure were used [15]. The same applies to the $CO_2$ transport route; here, too, leakages and unforeseen risks are feared by the population [15]. In addition, several stakeholders in Germany expected so-called spillover effects, which occur when already existing rejections of CO pipelines are transferred to $CO_2$ pipelines on the grounds that these transport options are not sufficiently differentiated in society [15].

In this context, the fear of a lack of acceptance of responsibility on the part of politics and industry [71,77] and the societal desire to avoid uncertainties are mentioned [30], especially when it comes to long-term monitoring of $CO_2$ infrastructure, which is primarily intended to ensure the protection of future generations [71,73]. In addition to health risks from the use of iCCS, ecological risks were also mentioned in the analyzed articles [15,75,76], which can have an unfavorable impact on acceptance. For example, interventions in the ecological system through the construction of new $CO_2$ infrastructure can permanently endanger the environment [15]. In addition, one study expressed fears about the possible effects of stored $CO_2$ in the seabed [73], which could, for example, affect the fauna and flora of nearby coastal regions and lead to catastrophic consequences there [71]. At the same time, the use of iCCS technologies was interpreted as a standstill for other climate protection measures in industry that would lead to lock-in effects of unsustainable corporate practices [73]. However, the results on the perception of iCCS technologies are partly contradictory; on the other hand, there is apparently the concern that without their use, no adequate emission reductions for the climate can be achieved by energy-intensive industries [62] (which can ultimately be seen as an advantage for iCCS).

In addition to these societal risks, the studies also mentioned some personal risks that may be decisive with regard to the perception of iCCS. These include, in particular, the previously mentioned perceived health risks, which could lead to a strong rejection of iCCS technologies, especially on the part of the local population [13,30,71]. Personal risks may also be perceived in conjunction with the economic factors of iCCS. For example, the results of the analyzed studies illustrate that the factor of employment can be perceived as both a personal risk and a benefit [14,65] for people in a region in the context of iCCS. For example, one study expressed concerns that iCCS may impose costs that are then offset by, for example, lower employment levels in iCCS operations. On the other hand, the introduction of the technologies could create new areas of work and if steps were taken to retrain and employ industrial workers within the iCCS sector, this would be a benefit [71]. However, there has been an equal concern that there may be inflation of products through use with iCCS and in the long run this effect will contribute to industrial companies becoming uncompetitive in the global market and may lead to local plant closures [65].

### 3.2.3. Preferences/Values

In the context of the studies analyzed, a variety of values and attitudes were explained that can have an influence on the acceptance of iCCS. These broadly include cultural identity, the closely related moral concepts of a society, environmental awareness, the perceived influence of iCCS on people's living conditions and attitudes toward technological developments and industry.

According to the study by [13], nationality is the strongest predictor of support for iCCS. Closely related to nationality is the cultural identity of a country. Thus, a study explained that compensation services to communities [74] must take into account the cultural as well as the social context [14,30,62]. Here, it is especially important that sacred values such as human safety are not mixed against a secular value, for example, by accommodating a hazardous facility in exchange for monetary compensation [74]. Certain normative ideas and moral values are also obviously advantageous for the development of a positive attitude towards iCCS [63,76]. Insofar as the use of iCCS can compensate for possible inequalities in society [65], for example, by allowing regions with a high proportion of energy-intensive industries to hold on to their economies to some extent or to operate them in a climate-friendly manner through iCCS, this represents an advantage for the perception of iCCS [64]. However, such perspectives do not go hand in hand with the moral notion that iCCS is interpreted as an intrusion into the subsurface "wilderness" or that BECCS is morally indefensible due to the still unclear availability of biomass, as stated in [71]. A view that, according to [71], occurs among those with strongly ecological values. According to [71], iCCS can only contribute to justice in society where a common understanding of cultural, natural and socio-economic systems prevails.

The influence of environmental awareness on the acceptance of iCCS is still evaluated very differently. Thus, ref. [13] clarify that a high environmental awareness can lead to a low acceptance of iCCS as the technology is seen as less important for coping with climate change than other technological options [63]. Whereas BECCS technologies seem to get a better rating in [71] compared to $CO_2$ capture from further industrial processes (here certain views of environmental awareness do not seem to be in conflict with the moral risks of BECCS mentioned above). Either way, BECCS is obviously viewed positively here because it is more likely to be associated with natural processes through the use of biomass [16]. However, if iCCS technologies are placed in the larger context of addressing climate change, where the technologies are embedded as part of an overall strategy to reduce $CO_2$, their perception as an environmentally conscious technology may change if necessary [13,65]. Here, the urgency to address climate change postulated in recent years seems to have become a helpful vehicle for improving society's perception of iCCS technologies [63]. Another step towards valuing iCCS as an environmental technology focuses on the perception of $CO_2$ as a significant resource [64] rather than a waste product (see Section 3.2.1) or iCCS as a socially desired argument to support energy-intensive industries in the context of political decarbonization intentions [53].

It remains open whether, far from being environmentally conscious, people can develop a positive perception of iCCS out of a certain technological affinity. The authors of [30] present a study in which people with a positive attitude toward gas infrastructure development are more supportive of iCCS than people without this attitude. In addition to environmental awareness and technological affinity, the perceived impact of iCCS on people's concrete living conditions is also likely to be significant in assessing acceptance [68]. For example, results from a focus group [71] illustrate people's fears that a life based on the renewable energy technology system may be very regimented and "robotic" and that this development may negatively affect previously valued lifestyles. In light of these considerations, the use of iCCS technologies is evaluated in a different context; in which through them traditional ways of life can be maintained for longer, which is evaluated as quite positive [70]. The authors of [13] also found in their study that people with energy-intensive lifestyles were more likely to prefer iCCS than others because they too could maintain their lifestyles while not being accused of promoting climate change.

The general attitude of the population toward the industry could also be an indicator for the future acceptance of iCCS. This is an aspect that will be discussed in more detail in the following section, as it is very closely linked to questions of the regional affiliation of the public.

### 3.2.4. Regional Factors

In this section, we will focus on the factors that can exclusively determine the regional characteristics and conditions for the development of iCCS acceptance (independent of other factors such as trust, knowledge, and communication, which can also influence the regional perception of iCCS). These factors on regional specificity include the specific history of an area and the regional perception of iCCS technologies in the context of other developments, such as the economic activities and geological conditions of the region.

The results of the studies analyzed suggest that despite the processes of deindustrialization in advanced capitalist economies, deeply rooted cultural narratives of industrial modernity and manufacturing employment remain powerful markers of identity and social progress [64,71]. In regions with an industrial heritage, where the local public feels connected to industry, this identity is particularly high [74]. Regional populations appreciate it when industrial actors inform them and involve them in their activities and plans to give them a sense of belonging and identity [66,74]. It is becoming apparent that people in such regions are concerned that these industries remain fully intact and are becoming sustainable [13,62].

Ref. [14] contribute to this thesis, for example, with the study of Teesside (UK). Teesside is a conurbation with a strong industrial base that residents rely on. Ref. [74] also assume that people in such regions are more positive about iCCS development than people who are less rooted in their industrial heritage. For example [66], describes that the Norcem industry began producing cement as early as 1919 and quickly became a major player in the economic life of the region. Ref. [64] emphasize the aspect of habituation. If people are used to industrial activities, especially when industry has operated in the area for decades, this has a positive effect on trust towards local industry and politics. For example, residents in northern regions are also accustomed to transporting products that are considered more dangerous than $CO_2$, such as ammonia.

Ref. [13] assume that areas where iCCS plants are likely to be built are typically those locations where (analogous) industry already exists. Subjective familiarity with such an industry could also serve to reduce the perceived risks associated with new infrastructure, leading to greater acceptance (or tolerance) of iCCS within regions. Fundamentally, according to [74], there is a need to understand local social realities, such as understanding what a particular place means to the local public, as well as how iCCS technology can impact this meaning at an early stage of the projects.

However, refs. [15,30,67] also emphasize that past economic activities, for example, when coal mines are present in the region or there have been incidents with health impacts for local residents, can have a lasting negative effect on the implementation of new projects. For example, the explosion of a gas pipeline in Belgium in 2004 increased public concern about the perceived reliability of $CO_2$ transport [30] (see also [15] regarding the CO pipeline in Section 3.2.2).

Another crucial factor for the regional acceptance of iCCS seems to be the specific perception of actors and issues related to a (possible) project. Ref. [74] suggests that this debate is also in the literature on the so-called social license to operate (SLO): "SLO refers to the informal permission granted to industry by the local community and wider society to develop a technology; in the context of CCS, SLO has been recognized as very preliminary and fragile". The following factors are summarized for achieving an SLO by [74] and are supplemented here by the results of other studies:

- Weighing the costs and benefits to the community, based on the particular characteristics of the project (see also [13]). Here, the ability of iCCS to protect jobs was identified as one of the key benefits. These benefits can be felt even more strongly for iCCS as it

both protects employment in existing industries and provides infrastructure that can attract new investment and employment opportunities [13,66,71];

- Creation of socio-political legitimacy; that is, whether an industry and all other (interest) groups act fairly, respect local lifestyles, and, in sum, the community plays a role and is involved (see also [13]). This can also include industry engagement with the local public, which is seen as the "key vehicle for achieving social license" by [81]. Part of this engagement can be compensation measures offered to the community [74];
- Creation of interactional trust; in which all participants engage in a mutual dialogue (in relation to communication, compare also Section 3.2.7);
- Establishing an institutionalized trust in which a lasting relationship with community representatives is established, taking into account mutual interests. This dialogue also includes the industry's ongoing efforts to address environmental challenges, including iCCS—see also [64].

In addition to the factors already mentioned, the studies identified further aspects that may have an influence on the regional acceptance of iCCS; these include the specific economic situation and the geological conditions of a region. These have already been discussed in more detail in Section 3.2.1 on the perceived benefits of iCCS and will not be repeated here.

### 3.2.5. Trust

In almost all analyzed studies (n = 23), the topic "trust" was treated as a crucial acceptance factor for iCCS. Ref. [74] conclude that research indicates that trust in developers and other stakeholders is a critical factor influencing public response to a development such as iCCS as a whole, as well as at the community level. Within the studies analyzed, the trust factor is predominantly discussed in the context of regional processes and stakeholders on iCCS. Some stakeholder groups enjoy more trust among the population than others. These groups include in particular (environmental) non-governmental organizations (ENGOs) and local stakeholders, for example politicians and investors, who are considered to represent local and civic interests [15]. These groups of people are thus seen as having a certain degree of integrality. Whereas [62] notes that in the context of a focus groups in Wales (United Kingdom), a distrust of both a major steel producer and the government at all levels was mentioned based on a lack of integrity and competence. According to [14], perceptions of trust in key institutions depend on the track record of those institutions in managing past industrial processes.

Local authorities seem to have a special role to play here in developing a deeper commitment, as they can act as facilitators for the deployment of iCCS [65]. The importance of the position of the municipality towards CCS projects has been shown in previous studies. In Barendrecht in the Netherlands, the local government rejected a proposed CCS project because they feared negative impacts on public health and a decline in property values [64]. Accordingly, it is important that the community, including the people who live there, feel that the continued efforts of industry to build technology like iCCS is also directed toward solutions to environmental challenges [64]. This is where community familiarity with industry relevant to CCS implementation may also be important [64]. Moreover, ref. [13] argues that subjective familiarity with such an industry may serve to reduce the perceived risks associated with new infrastructure, leading to greater acceptance of iCCS within the intended communities.

At the same time, gaining public trust is an extremely lengthy and labor-intensive process that is highly dependent on experience in the interaction between laypersons and project stakeholders [30]. It is also important to avoid violating trust as much as possible, as it can be difficult to rebuild and can also cause negative spillover effects on perceptions of other technologies and projects [14]. Distrust can have an effect in different areas, on the one hand with regard to the competence of the responsible persons (competence-based distrust), especially when it comes to the implementation of a complex infrastructure project such as iCCS technology [62]. On the other hand, distrust can also relate to procedural fairness in

the participation process (integrity-based distrust [62]; compare also the comments on socio-political legitimacy in Section 3.2.4). According to [74], without a more comprehensive public involvement strategy, the question remains whether this is sufficient to build a sense of trust towards the developer.

### 3.2.6. Knowledge/Awareness

As expected, none of the studies analyzed provide any information on what the state of public perception and knowledge of iCCS technologies is. However, the results of [13] show that public awareness of CCS (without concreteness to iCCS) remains low (here for Canada, the Netherlands, Norway, the UK and the US) and this result is also in line with previous research. However, in deciding whether to accept or reject CCS, the general level of knowledge and awareness plays an important role, as illustrated by the presentations from Tcvetkov's literature review on CCS [30]. Stakeholders interviewed by [15] in the ELEGANCY project rate public knowledge about CCS as rather low and perceive that iCCS technologies are not yet present in the current public discussion due to low market penetration. The results of [61] in the context of an experiment suggest that iCCS is viewed more positively by those who claim to have more knowledge about iCCS and that they are also likely to show a higher interest in the technology. Additionally, ref. [41] found that higher information levels can fundamentally change the evaluation of $CO_2$ capture options (for example air capture or from chemical plants).

The study [64] emphasizes that the local population in Porsgrunn (Norway) is not only used to industrial activities, but is also likely to have concrete experience with iCCS activities. There is a sense that the local population is positive about the proactive approach to managing $CO_2$ emissions, and this assumes that there is some level of knowledge about iCCS locally. Beyond this level of knowledge about iCCS, ref. [77] clarified that industries also have an interest in iCCS technologies becoming more widely known. For example, to market BECCS, public knowledge of low-carbon technologies is a possible positive aspect. The reasoning is that customer demands for negative emissions make investment decisions easier for industries because they can integrate iCCS technologies as part of their sustainability strategy. According to [65], however, even key stakeholders such as trade unions and environmental organizations lack evidence-based information on the iCCS capabilities of carbon-intensive industries. Ref. [73] also assumes that environmental organizations (related to Europe) lack the necessary resources to acquire knowledge about different iCCS technology options in detail. This lack of capacity also contributes to the apparent lack of official positions on issues such as iCCS until 2018 [73].

Beyond just awareness and knowledge of iCCS, the studies address the need for contextual knowledge. For example, ref. [72] suspects that there will be a more positive perception of iCCS as people become more aware of their individual climate impacts. Thus, some of the stakeholders interviewed in the study of [15] also see a general lack of societal acceptance regarding energy technologies and large-scale infrastructure, attributed in part to a lack of knowledge. Perception of global warming issues, understanding of the role of humans in this process, and developing an objective view of the prospects of low-carbon technologies, including CCS, depend on the education of respondents [26,30]. Therefore, implementation of an educational strategy for sustainable development should be considered, which starts at school and could be part of a national "green" policy. Ref. [71] clarified in their study that with the level of knowledge about iCCS and the integration of the technologies into a higher-level thematic context, the initially perceived assessment of iCCS can change once again. If iCCS is initially interpreted as a potential threat to natural systems, subsequent presentations and scenario discussions led to a gradual shift in how participants interpreted iCCS. Similarly, ref. [62] clarifies that participants in two focus groups on the Port Talbot steel mill development acquired contextual knowledge to evaluate iCCS. For example, they express concerns that if iCCS makes steel more expensive, the Welsh steel industry could lose out to foreign competitors who continue to produce emissions-intensive steel at the lowest price. If nothing else, these findings illustrate

that awareness of iCCS does not immediately predict public acceptance of a project [30]. Ref. [66] also note that regardless of the depth of their insight and knowledge, people will acquire subjective perceptions about iCCS. Ref. [30] sees consolidating government, industry and NGO efforts as one of the key challenges to improving public perceptions of CCS.

### 3.2.7. Communication/Participation

The discussion of CCS communication and participation in the articles analyzed is extensive and is therefore presented in the form of a table (Table 3). Ref. [68] suggests that the CCS community is generally aware of the range of factors that influence public engagement. Whether this range changes significantly for communication about iCCS cannot be adequately answered using the available results. Ref. [74] illustrates that effective public engagement will be key to successful iCCS implementation. With this comes the need to further explore how to most effectively engage with the local public.

**Table 3.** Overview of the acceptance factor "communication/participation" of iCCS (who/what/how).

| **Who should communicate?** |
| --- |
| Persons of trustPersons within the scope of their respective expertise |
| Qualified project team |
| Entire community of interest (to be defined on a case-by-case basis) |
| Inclusion of new players, e.g., business and trade associations, companies along the entire value chain |
| **What should be communicated?** |
| iCCS narrative embedded in the overall context of sustainability |
| Urgency to combat climate change |
| Framing of iCCS as environmental technology (where there is no alternative) |
| Discussion of alternative technologies |
| Integration into norms and values of society |
| Costs in the context of the overall energy transition |
| Economic advantages and disadvantages |
| Set economic consequences in relation to ecological ones |
| Infrastructure challenges/use of existing infrastructure |
| Presentation of project experiences incl. risk analyses |
| Integration into current political context |
| Liabilities/standards/regulatory framework/securityRole of iCCS for global economy/international cooperation |
| **How to communicate?** |
| Develop an empowerment and communication strategy and plan |
| Take into account the main principles of public participation |
| Meaningful voice during decision-making processes |
| Establish continuity in communication |
| Fairness/greatest possible transparency/inclusion of all/neutral/clear/high quality |
| Creation of problem-oriented knowledge, e.g., FCDP |
| Include local needs and contexts/site characterization. |
| Consider community compensation |
| Use of classic media, such as brochures, local media |
| Facilitate face to face exchange, e.g., local activities and events |
| Use of digital media |

The chosen order of the factors does not represent a weighting.

In this context, it seems important to mention again the aspect of [74], which emphasizes a certain flexibility in dealing with iCCS projects, as specific concerns and needs may change over time in different regions. Here, regular adjustments of the implementation strategy of iCCS projects have to be taken into account.

### 3.2.8. Socio-Demographic Factors

The analysis of the influence of socio-demographic factors on the acceptance of iCCS from the available studies does not reveal any meaningful trend. According to [67], for example, the acceptance of iCCS among women is about three times higher than among men (in selected European countries). Additionally, according to [13], men (as well as older people and people with high incomes) showed lower support for iCCS (but only after reading the message on CCS and possible lifestyle change). In contrast, ref. [30] presents findings in which men show more tolerant perceptions of CCS risks when the economic potential is present, while women are more concerned about safety. Additionally, as mentioned earlier in the context of a country's cultural identity (see Section 3.2.3), nationality represents the strongest predictor of support for iCCS [13].

All other results on the influence of the socio-demographic factor do not explicitly refer to iCCS technologies and therefore do not find any further explanation here.

### 3.2.9. Perceived Differences between CCS and iCCS

In the following, the question is addressed whether significant differences between the acceptance of CCS from fossil-fired power generation plants and the acceptance of iCCS from industrial processes can be derived from the results of the analyzed studies. There are a number of initial results on this, but they target different technology pathways and are therefore hardly comparable. First, ref. [30] suggests that CCS technologies received general support from respondents in a survey, but when it comes to specific options for implementation, for example as part of gas and coal-fired power plants, initial public preferences may be negated. Additionally, according to [71], focus group participants articulate more positive visions for iCCS and BECCS than for coal CCS. They affirm support for growth through iCCS in manufacturing industries, as this is highly desired by society. Additionally, ref. [15] assume that iCCS will have higher social acceptance than CCS. Beyond this more economic aspect, ref. [68] represents the need to significantly broaden the iCCS discussion to include heavy industry and processes outside of power generation. This was seen as necessary to counter the traditional arguments of environmental groups that reject CCS because of its ability to re-generate electricity. In addition, initial studies compare the acceptance of iCCS with the acceptance of gas-fired power plants. For example, ref. [16] show in their experiment that BECCS plants receive higher approval than those using conventional gas. Interestingly, as perceptions of BECCS improve, so does the willingness of one's community to accept $CO_2$ storage. Ref. [17] also found that large-scale plants converting gas to hydrogen (H2) with CCS tend to be viewed negatively by most respondents. Basically, ref. [71] assumes that fossil CCS is considered unacceptable by the local population, while other CCS options, like iCCS, remain feasible.

### 3.2.10. Evaluation of iCCS for Different Process Steps

iCCS technologies encompass many different technological concepts and potential target applications. The results presented below are intended to illustrate the acceptance of iCCS along the stages of different value chains and the underlying factors. It should be mentioned at the outset that the studies analyzed did not examine in detail the possible effect of the technical feasibility of different iCCS technologies on iCCS acceptance.

The following findings are available on the $CO_2$ source and the capture process step:

- BECCS: as briefly indicated before, BECCS is preferred to fossil-based CCS. According to [76], the technological approach has reached a stage of normalization in the debate, at least in the scientific discourse, after several years of intense criticism, and has become a self-evident aspect of climate change discourse. Especially for countries with a strongly biomass-based economy, such as Finland, BECCS seems to generate benefits [75]. With reference to [71], CCS was seen as a more intuitive and natural process when linked to managed forestry and the carbon cycle. Similarly, ref. [41] presents the use of biogas plants as a source of $CO_2$ as a promising option for industry and policy makers to achieve a socially acceptable form of carbon capture. Environmental

organizations such as Greenpeace and Biofuelwatch disagree here, according to [76], emphasizing problems with agricultural production and water scarcity in the context of BECCS. This aspect is also critically addressed in the Convention on Biological Diversity from 2019 [82]. This is because significant negative impacts on biodiversity and food security are expected as a result of the extensive land use changes caused by the consistent use of bioenergy, including BECCS. It remains to be seen what effect this position can have in terms of shaping public opinion. However, ref. [13] assume that BECCS is more supported than shale gas, underground coal gasification, and the application of CCS in heavy industry.

- Post-combustion capture: while the process can be retrofitted into existing energy infrastructure, it does not promise economic feasibility due to low efficiency and increases the need for fossil fuels, thus having a comparatively high environmental impact. For these reasons, the process is generally not considered beneficial from the perspective of interviewed stakeholders [69]. In contrast to oxy-fuel technology, post-combustion requires larger constructional measures and entails a visible and significant change to the existing plant. Therefore, acceptance-relevant aspects may occur due to construction sites and changes in the landscape [15].

- Direct air capture (DAC): according to [41], capturing $CO_2$ from ambient air is not an accepted option among the public, especially when detailed information on efficiency and energy requirements is available.

- $CO_2$ capture from chemical plants: the results of a study by [41] show that providing technically correct and comprehensible information has the potential to completely revise previous negative opinions of study participants. The prerequisite is that it is explained transparently that the capture of $CO_2$ from a chemical plant is highly efficient and has a lower environmental impact compared to other alternatives. Initially negative reactions can thus be transformed into positive acceptance ratings.

The following findings are available on the acceptance of the $CO_2$ transport process step:

- Rejection of $CO_2$ pipelines: Respondents' judgments in an experiment by [16] were most influenced by the pipeline factor, to a lesser extent by the plant factor, and least by the storage location factor (there are a variety of contrary results on this). However, people seem unwilling to live near a pipeline (respondents from Switzerland), although they would prefer a $CO_2$ pipeline to a gas pipeline. Field testing of geological storage in densely populated areas may therefore consider avoiding pipeline transport to increase the likelihood of public acceptance [13].

- Use of existing infrastructure: ref. [41] make clear in their study that $CO_2$ transport by truck and a mix of trucks and pipelines are not preferred by the participants. In particular, the negative ecological effects expected for the construction of new infrastructure packages are mentioned here. Instead, it is recommended to examine the potential of using the existing infrastructure for alternative fuel production. A further step would even be the avoidance of $CO_2$ transports by spatially linking $CO_2$ capture and fuel production—an option that should be examined in terms of acceptance.

The following findings are available on the acceptance of $CO_2$ use:

- Methanol production: according to [30], the most preferred way to use $CO_2$ is methanol production, while the CCS-EOR process chain is perceived as one of the worst alternatives, second only to CCS without the link to the beneficial use of $CO_2$.

- Chemical looping and $CO_2$ removal from calcination processes: these have shown potential according to [69] in the study area of Finland, especially in small CCU applications and in some cases also in CHP production. Opportunities to recycle the captured carbon could help solve the economic feasibility problem due to lower transportation and storage costs and potential revenue from recycling. Whether optimizing economic feasibility may also have an effect on public perception is not addressed.

- $CO_2$-based fuel production: ref. [41] make clear that the public is less interested in the process step of $CO_2$-based fuel production and efficiency improvements in chemical production, but rather in the processes of $CO_2$ capture and transport.
- $H_2$/CCS value chain: ref. [14] represent that the H2 part of this joint value chain is more socially accepted than the CCS part. Nevertheless, the type of $H_2$ (green, blue, conventional) is also estimated to be relevant for acceptance. They also hypothesize that only established larger industries can address these infrastructure issues, but that the trust on the ground, where the (re)construction of the infrastructure takes place, is more likely to be given to local stakeholders.

The following findings are available on the acceptability of $CO_2$ storage in conjunction with iCCS:

- Onshore storage: ref. [16] suggest avoiding the NIMBY (not in my backyard) effect in field trials of $CO_2$ storage using BECCS as the $CO_2$ source. It is likely that the source of the $CO_2$ is critical to the acceptance of the storage site.
- Offshore storage: Haug's results show that the possibility of the offshore storage of $CO_2$ could be a clear advantage for the Nordic regions for the establishment of an iCCS economy [64]. As an example, the municipality of Porsgrunn in Norway, whose positive attitude towards existing and potential iCCS activities may result from the option of offshore storage, should be mentioned once again. The Sleipner project in the North Sea was also realized without much public controversy, and ref. [64] suggest that this could also be a result of the offshore location. In sum, the off-shore option could be a great advantage for the Nordic region, but it is important to note that it must also gain the consent of the stakeholders in the use of the sea and that there is no guarantee of acceptance if these stakeholders are neglected [64].
- Geological and infrastructural prerequisites: Countries with an interest in establishing an iCCS economy should carefully examine their geological prerequisites. According to [75], $CO_2$ storage is an open question in Finland, as the country lacks potential geological formations for it, which also underscores the importance and cost of $CO_2$ transport [75]. Russia, on the other hand, has extensive area and therefore allows $CO_2$ storage at a considerable distance from industrial centers and residential areas, which could potentially weaken stakeholder opposition to the projects [70]. Another option, he said, is to look at reusing existing infrastructure for $CO_2$ storage, as proposed in the Acorn project. Significant cost savings can be achieved through this approach, and this also represents a societal approach to enable broader CCS deployment [78]. For example, existing $CO_2$ transport and storage infrastructure could be shared by multiple capture projects to maximize value, simplify investment decisions, share operating costs, and thus reduce development costs.

Finally, it should be summed up here that several studies consider the acceptance of iCCS along the different process steps and value creation stages to be possible. An important approach to developing iCCS acceptance, initially primarily from an economic perspective, is the pursuit of a cluster and network approach [14,41,62,64,67,78], which is already emerging as a trend in practice (see Section 3.2.1 for a more detailed discussion).

### 3.2.11. Regulatory/Political Aspects

This literature review also noted circumstantial evidence suggesting that a lack of regulatory frameworks, political support, and missing or complex approval processes may influence iCCS adoption.

The findings highlight a fundamental need for strong regulation and policy on iCCS, both to leverage the skills and experience of the private sector and to maintain the common good and public interest [65]. For example, a UK opinion poll cited by [62] found that a majority (74%) of adults support policies to regulate heavy industry to ensure emissions reductions in the sector. Focus group participants from a region of Scotland that has historically been closely associated with energy-intensive industry (Port Talbot steelworks) assume that there will be stricter emissions legislation for these industries in the long

term, and therefore refer to iCCS as an "inevitable" option [62]. This would imply that expectations of stricter emissions legislation in the future from national and EU levels alone can convince people that iCCS is inevitable in the future. On the other hand, the participants of this study also valued the European Union as an important partner for the implementation of iCCS technologies [60], especially by providing the necessary funding. In this context, ref. [69] also mention the funding for the development of the necessary $CO_2$ transport infrastructure.

Ref. [30] go one step further and assume that an important factor for further iCCS development is international cooperation. On the one hand, so that individual countries can embed and position their iCCS policies internationally [14], and on the other hand, international cooperation would make it possible to combine national efforts, create favorable conditions for project proposals and adopt successful experiences of other countries. Thus, it would be necessary to create a political context that can strengthen public trust due to the importance of collaborative decision-making [30]. Local and regional networks alone would be insufficient to influence national policy [14,63]. In addition, ref. [65] describe that there would be limited public communication of an iCCS project proposal if political uncertainties prevail. For this, it is also important to have political long-term strategies that create reliability, for example, regarding BECCS technology and its integration into the European Union Emissions Trading Scheme (ETS-EU) [77]. This integration would be important for Finland, for example. Without BECCS, it would be challenging to meet emissions targets, but with BECCS, Finland could gain benefits by saving and trading emissions allowances [75]. The need for ETS-EU was frequently mentioned in the analyzed studies, but mostly by industrial actors and other experts [69,75,77].

## 4. Discussion

The present study is the first literature review to address the acceptance of iCCS. The objective of this study was fourfold. Firstly, it is examined to what extent the acceptance of iCCS is already being empirically investigated. Secondly, an analytical framework is proposed in order to systematically review the existing literature. Thirdly, based on the review, factors influencing the acceptance of iCCS are identified and discussed. Fourthly, results for the acceptance of iCCS are compared to CCS, highlighting some important differences between the two areas of application.

First, the results show that there is still only limited research on the acceptance of iCCS. Between 2012 and 2020, 25 scientific articles were published on the subject, with very different and incomparable methodological tools and research questions.

Secondly, during the evaluation process, it became apparent that the analytical framework transferred from CCS acceptance research, with its well-established dimensions (cf. Table 1), was sufficient to systematically gather the results from the articles. The research findings of the analyzed articles could be assigned to one or more dimensions, such as findings on local aspects (as suggested by Table A1 in the Appendix A, see column "Important statement related to iCCS"). Influencing variables that emerged in the analyzed articles and initially deviated from the established factors for CCS acceptance research (for example, the employment factor) could be assigned to the existing dimensions by the author during the evaluation. Accordingly, no further factors were inductively added to the analytical framework established in Section 2. As a result, many factors explaining the acceptance of CCS seem to be decisive for the acceptance of iCCS as well. However, it became apparent that the weighting and the expressions of acceptance factors to iCCS appears to vary compared to CCS, as shown in the following. Moreover, only tentative trends for the acceptance of iCCS can be derived from the studies analyzed. It remains unclear whether iCCS applications are more likely to be accepted or rejected by society in the future. Moreover, from a scientific point of view, a methodological concept for analyzing iCCS acceptance is still lacking, even though the factors considered here already provide a good starting point for operationalizing the research subject. Given the wide

range of technological options and the resulting societal implications, this task also appears to be non-trivial.

The discussion of objectives 3 (factors influencing iCCS) and 4 (differences of CCS and iCCS) of this content analysis are now discussed in conjunction.

More specifically, acceptance at the regional level, for example, appears to depend even more significantly on the perceived societal benefits that people associate with iCCS. The potential to maintain and increase local employment through the use of iCCS applications was frequently mentioned [13,64,66,71]. This represents a difference from the debate in perceptions of the societal benefits of CCS. In sum, it appears as if the population expects the safeguarding or even increasing of economic performance in their local environment with the use of iCCS. Previous research findings illustrate that societal benefits have either the same or slightly higher explanatory power for CCS acceptance than societal risks [31,35,47,83] (see Table 1). Whether this is also valid for the acceptance of iCCS remains to be investigated.

What is clear is that both factors will also be very significant in the context of iCCS. Subjectively perceived risk associated with $CO_2$ storage has been a crucial factor in explaining local and regional resistance in the context of CCS technologies [11,16,44]. It is different from factual risk in this regard as [53] illustrated with their approach to misconceptions. The fact is that $CO_2$ pipelines are state of the art and have been operating in the United States for example since the 1970s. Additionally, no significant research and development budgets are being spent on $CO_2$ transport and the associated potential risks worldwide. In contrast, geological storage of $CO_2$ has been the subject of intensive research and development work internationally for many years, even though $CO_2$ storage is already being successfully operated in many countries [84]. Here, the exploration methods for $CO_2$ storage, the procedures for storage monitoring, the competition with other storage utilization options, the impact on geothermal energy utilization, and the theoretically possible effects on drinking water supplies are often the subject of interest [85]. In sum, the question is not so much whether $CO_2$ storage is fundamentally possible, but under what conditions it is as safe as possible. Besides these science-based facts of technical and environmental aspects, the subjectively perceived risk factor will be important in the context of iCCS acceptance, as many of the studies analyzed have made clear [13–15,30,62,65,68,71,73–77]. It seems that in this context the aspect of fair distribution of risks and benefits has to be more in focus than in the context of the CCS debate. If, in the future, the benefits associated with the use of iCCS are perceived by the population primarily at the global level in the context of climate protection and the local population gains the impression that, in contrast, they are more likely to be confronted with the disadvantages of iCCS applications, this would probably be a barrier to the development of acceptance. In relation to the perception of an equitable distribution of risks and benefits, the explicit understanding of the benefits associated with iCCS for a region therefore seems to be of importance. This starting point of an unequal distribution of risks and benefits in the context of the future deployment of iCCS offers a possible field of action, both for research and for the implementation of practical iCCS projects. The previous research approaches of possible compensation benefits in the context of CCS will be examined here for their transferability and applicability.

Furthermore, the factor trust, which was evaluated in most studies as an important tipping point for or against the acceptance of iCCS (see [13–15,30,62,64,65,74]), should be further investigated. It became clear that in the development of local iCCS projects, trust in the stakeholders involved becomes especially important when it comes to large infrastructure measures related to $CO_2$ transport [13,16,41,62]. It seems that the process step of transport has become critical to the CCS debate, even though the negative sign in the assessment of $CO_2$ pipelines does not seem to have changed. With respect to transport infrastructure, more knowledge is still needed on the acceptance of iCCS. It is unclear whether, for example, the "joint" use of infrastructure or the use of existing infrastructure by industry clusters or hubs leads to an improvement in the acceptance of iCCS. Another research question could be whether the $CO_2$ source has an influence on the acceptance

of $CO_2$ transport, for example if the source is associated with an industry that is deeply rooted in the local society and contributes to its identity.

In this context, the role of framing or a possible narrative for iCCS (and also the greenhouse gas $CO_2$) implementation should also be further explored. The studies have illustrated that framing iCCS as a climate change mitigation technology can lead to both positive and negative acceptance tendencies [13,16,53,63–65,71]. There does not seem to be a determination yet as to whether or not iCCS technologies are perceived by society as a climate change technology. This framing was hardly conceivable in the context of the CCS debate since there were sufficient technological alternatives for sustainable generation of electricity through the use of renewable energy technologies. Anyway, it is clear that iCCS operators would benefit from such "green" framing of iCCS applications, especially in marketing potential products along the value chain. This framing approach, based on a rather economically oriented marketing strategy, would certainly fall short. Ultimately, there is an obvious need for a more overarching narrative that takes into account both the aspect of sustainability and the reduction of $CO_2$, as well as economic issues that not only affect individual technology paths, but in sum relate to the economic viability of an entire region. As a consequence, this would mean embedding iCCS in a discourse around sustainable structural change. After all, regions with energy-intensive economic sectors are particularly affected by the challenge of structural change.

The articles analyzed have also made it clear that the factors of social "values and attitudes" can be significant for the acceptance of iCCS [13,14,30,62,64,65,71,74,76]. In this context, further research is particularly needed on the question of whether a certain environmental awareness has a positive or negative influence on the perception of iCCS. Compared to the CCS context, the clarification of this research question seems to be much more complex due to the many different possible applications of iCCS. In the context of CCS acceptance, existing studies indicate that people with high environmental awareness tend to evaluate the technology negatively [49,86]. Some authors mentioned that, triggered by the Paris Agreement, the absolute urgency of the transformation to a sustainable economy and way of life has now arrived in the perception of society. In light of this urgency, the evaluation of iCCS could also be developed in a more positive direction [63,65]. Again, there are only assumptions and no evidence-based findings yet. Interestingly, in one study, this urgency emerged as a driver of iCCS acceptance. This happens when this urgency is interpreted by society as a threat to their current lifestyles and cherished habits for everyday life, and iCCS is perceived as an option to hold on to these habits without regret [71]. This approach could also be a starting point for new research questions on the acceptance of iCCS. In addition to this urgency, another aspect could influence the perception of iCCS in the future. For example, the Global Assessment Report on Biodiversity and Ecosystem Services (IPBES) [87] does not exclude CCS (and thus iCCS) as a measure to mitigate negative impacts on biodiversity (see Glossary). Consequently, iCCS could be considered not only an option to reduce global $CO_2$ emissions, but also a generally accepted measure to avoid or limit potential negative impacts on biodiversity. If such a perception is perpetuated among individuals with a high level of environmental awareness, this aspect could be interpreted as an advantage for the use of iCCS and possibly have a positive impact on social acceptance. Whether this assumption is well-founded needs to be explored in future studies on the acceptance of iCCS.

## 5. Conclusions

The IEA [88] estimates that iCCS in the cement, iron and steel, and chemicals sectors will need to deliver around 28GtCO$_2$ of emission reductions between now and 2060 to meet the climate target of the Paris Agreement. To achieve these reduction goals globally, strategies for robust and timely market introduction of iCCS technologies need to be developed. For such a market introduction of iCCS, social acceptance is of particular importance in addition to technical-economic and environmental indicators, as the example of CCS has illustrated.

In the studies analyzed, a large number of indications for the design of a communication strategy were derived, largely on the basis of the findings from CCS acceptance research as well as on the basis of all the research on energy transformation (see Table 3). In view of the abovementioned abundance of requirements for such an iCCS communication, the question arises as to which institution is capable of organizing such a permanent and trust-based process and in which larger thematic context this communication can be embedded? This appears to be a difficult question to answer, especially against the background of often missing political strategies and the related regulatory frameworks on the national level. The present literature analysis shows on the one hand which starting points for the market introduction of iCCS exist so far from social science research for political and economic actors and on the other hand which research efforts are still required.

**Funding:** This research received no external funding.

**Data Availability Statement:** The articles used in the literature analysis can be accessed via the respective publishers (for more details, see the bibliography).

**Conflicts of Interest:** The author declares no conflict of interest.

## Appendix A

**Table A1.** Overview of the analyzed articles.

| First Author (Year of Publication) [Reference] | Method | Country | iCCS-Related Technology | Important Statement in Relation to iCCS |
|---|---|---|---|---|
| Alcalde et al. (2019) [78] | Evaluation of ACT Acorn findings and review of scholarly/industrial literature | UK | Complete iCCS value chain (Acorn Project) | Seven key elements for iCCS projects: Infrastructure reuse, storage development plan, low-carbon build-out options, full-chain development plan, policy support, just transition, public engagement, and knowledge exchange. |
| Aursland et al. (2019) [66] | Case study with local residents and Norcem employees (n = 15, face-to-face) | NO | $CO_2$ capture from the cement industry | Positive image of cement company conducive to acceptance, effects on local employment and environment perceived as benefits. However, also concern whether project affects local living conditions. |
| Boomsma et al. (2020) [74] | Literature review from academic literature (non-systematic, n = N/A) and publicly available documents (n = 25) | Academic literature: international; public documents (DE = 7, NL = 4, RO = 5, UK = 9) | No specific iCCS technique defined (focus on community compensation) | When implementing iCCS projects, it is important to understand local social conditions and examine what impact they have. Sites where the local public feels connected to the industry may be more positive about iCCS development. Compensation for communities needs to be integrated into broader public involvement strategies. |
| Broecks et al. (2016) [63] | Quantitative online survey representative for NL (n = 920) and discrete choice experiment | NL | No specific iCCS technique defined (=industrial applications) | "Industrial applications" is the most convincing pro-argument for CCS, followed by "dispose of $CO_2$ garbage", "safety of natural gas fields". Arguments on climate change are less convincing. |

**Table A1.** *Cont.*

| First Author (Year of Publication) [Reference] | Method | Country | iCCS-Related Technology | Important Statement in Relation to iCCS |
|---|---|---|---|---|
| de Best-Waldhober et al. (2012) [17] | Quantitative study (ICQ [1]) representative for NL (n = 971) | NL | Large plants where gas is converted into hydrogen with CCS | iCCS option rated lower compared to other energy production/mitigation options (except nuclear). |
| Dütschke et al. (2015) [61] | Quantitative online experimental survey design representative for DE (n = 1.672), assessment of 18 scenarios | DE | Industry and biomass power plant as $CO_2$ source | CCS scenarios that include either an energy-intensive industry or a biomass power plant as a source of $CO_2$ are perceived more positively than scenarios in which the $CO_2$ is captured from a coal-fired power plant. Rating of the respective $CO_2$ source as the strongest predictor. |
| Glanz et al. (2021) [15] | Qualitative explorative stakeholder interviews (n = 10) | DE | Hydrogen and carbon capture and storage infrastructure/ chain | Restricting the use of CCS for certain applications (industry, bioenergy) represent trade-offs that are supported by various stakeholder groups and offer a balance of environmental and economic arguments. Assumption: only large industries can address iCCS/H2 and its infrastructure challenges, but local trust is given to other stakeholders. |
| Gough et al. (2018) [14] | Mixed-methods approach: stakeholder interviews (n = 12) and two focus groups (n = 8 each group) with lay public | UK | iCCS with focus on $CO_2$ storage | Success of iCCS activities in a community dependent on social context, trust in key actors, track record of previous industrial processes. Hurdles related to procedural justice. |
| Haikola (2019) [76] | Qualitative analysis of (popular) science and news media from 2008—2018 (n= ca. 800) | International | BECCS | Scientific discussion about BECCS is becoming more neutral due to the time pressure to take action on climate protection. Debate moves away from the question of moral hazard and focuses instead on the need to act. |
| Haug et al. (2016) [64] | Interviews with municipalities (n = N/A [2]) and literature review | DK, NO, SE | No specific iCCS technique defined | Communities can consider iCCS as an advantage for regional value creation. Positive evaluation if local population is used to industrial activities and has concrete iCCS experience. Potential for offshore storage in a region is evaluated as an advantage. |
| Ilinova et al. (2018) [70] | Case studies (n = N/A), stakeholder management tools, and a checklist method | International | No specific iCCS technique defined | Most attention in CCS project planning/implementation should be focused on industrial companies/investors, government and society. CCS projects are mostly local projects; however, they are implemented in the context of national and even international interests. Therefore, the circle of stakeholders is large and establishing a constructive dialogue with all proves to be a difficult task. |

**Table A1.** *Cont.*

| First Author (Year of Publication) [Reference] | Method | Country | iCCS-Related Technology | Important Statement in Relation to iCCS |
|---|---|---|---|---|
| Kashintseva et al. (2018) [65] | Empirical model based on representative online survey (n = 564) | CZ, DE, IT, NL, PL, SK, UK | No specific iCCS technique defined (iCCS products and technologies) | Increase of iCCS sites, including those in the neighboring regions and countries, leads to the increase of negative consumer attitudes to iCCS and renewable energy policies. NIMBY effect is considered relevant. |
| Kojo et al. (2017) [75] | Quantitative longitudinal analysis of newspaper articles from 1996–2015 (n = 282) | FI | No specific iCCS technique defined (pertains to BECCS) | Agenda setting of the media regarding CCS is strongly dependent on real plant projects and communication measures of industrial actors. iCCS actors are not yet involved in communication in Finland. Business models are missing, costs are overestimated, a debate specifically about possible international developments is missing. |
| Offermann-van Heek et al. (2020) [41] | Quantitative online survey representative for DE (n = 300) and best-/worst-case scenarios | DE | DAC, biogas and chemical plant | Capture and transport process step more relevant to public than further use of $CO_2$, use of existing infrastructure conducive to acceptance, $CO_2$ use from BECCS and chemical plants viewed positively, DAC not an accepted option. |
| Pihkola et al. (2017) [69] | PESTEL [3] framework (analysis macro-environment of industries), stakeholder interviews (n = 12) from 2011–2012, media analyses (n = N/A), literature reviews | FI | No specific iCCS technique defined (pertains to BECCS) | iCCS needs a regulatory framework and political support, especially for the development of infrastructure. More systematic and differentiated consideration of iCCS applications is required for Finland. BECCS/CCU is seen as an opportunity for iCCS due to the central role of the Finnish energy-intensive industry. |
| Rodriguez et al. (2020) [77] | Qualitative inductive interviews with company representatives (n = 20) | FI, SE | BECCS | BECCS is technically feasible; what remains unclear is who will create a financially viable business case and establish supporting policies, as well as who will build the necessary transportation and storage infrastructure. In addition, customer requirements for negative emissions are still lacking. |
| Serdoner (2019) [73] | Qualitative interviews (n = 3) with representatives of EU environmental organizations, analysis of their public relations activities and literature review | EU | No specific iCCS technique defined | Positions of ENGOs operating in Europe on iCCS are closely related to previous debates on the application of the same technology in the power sector. Previous experience has led ENGO to approach the technology with skepticism and caution. They are either neutral toward iCCS or opposed to it. |

**Table A1.** *Cont.*

| First Author (Year of Publication) [Reference] | Method | Country | iCCS-Related Technology | Important Statement in Relation to iCCS |
|---|---|---|---|---|
| Swennenhuis et al. (2020) [65] | In-depth semi-structured interviews (n = 25) with regional stakeholders and workshops (UK) | NO, NL, UK | No specific iCCS technique defined | Narrative that iCCS is deployed for benefit of citizens/communities/workers and not in support of private sector, policy that leverages private sector capabilities without setting aside the public interest, need for deeper engagement with local governments that act as facilitators for iCCS deployment. |
| Tcvetkov et al. (2019) [30] | Literature review from 2002–2018 (n = 135) | international | No specific iCCS technique defined | Development of a regulatory framework to control the industry, important for public trust.Public preferences regarding capture plants are explained by problems with existing energy infrastructure. Public trust in environmental arguments of industry lower compared to NGOs, arguments of industry about economic aspects of project implementation are better perceived than by NGOs. |
| Thomas et al. (2018) [71] | Two qualitative deliberative workshops with local population (n= 12 each) | UK | Industrial CCS and BECCS | Depending on the context, iCCS may be perceived as a threat or a support to local social and economic interdependence. As a threat, for example, through costs that could harm employment in local industries, as a benefit through protecting and at the same time rejuvenating historical employment patterns through iCCS. |
| van Os (2018) [72] | Interview with Peter van Os | NL | Complete iCCS value chain (ALIGN CCUS Project) | Assumption that there will be a more positive perception of CCUS as the public becomes more aware of their individual impacts on climate. Uncertainties related to the cost of implementing CCUS, costs will decrease as implementation of CCUS technology progresses. |
| Wallquist et al. (2012) [16] | Online Experiment (n = 139) | CH | BECCS | $CO_2$ source decisive for acceptance of storage site, avoidance of $CO_2$ pipeline transport in densely populated areas, avoidance of the NIMBY effect through the use of BECCS. |
| Whitmarsh et al. (2019) [13] | International experimental online study (n = 5.406), national and local samples | CA, NL, NO, UK, US | No specific iCCS technique defined | Bioenergy with CCS is more supported, while shale gas, underground coal gasification, and heavy industry with CCS are less supported. Areas where CCS facilities are likely to be built are typically locations where (analogous) industry already exists. Subjective familiarity with this industry could serve to reduce perceived risks associated with new infrastructure. |

**Table A1.** *Cont.*

| First Author (Year of Publication) [Reference] | Method | Country | iCCS-Related Technology | Important Statement in Relation to iCCS |
|---|---|---|---|---|
| Williams et al. (2021) [62] | Two qualitatively designed focus groups with citizens (n = 11 and n = 10) | UK | iCCS in the steel industry | Community could endorse use of iCCS if developer/government collaborate from local to national level, provide transparent dialogue process that supports community trust in intent, integrity, and competence of implementing organizations. |
| Xenias et al. (2018) [68] | Mixed-methods approach: interviews (n = 13) and online survey (n = 99) with experts | Interviews: NO, NL, UK; Online survey: DE, NL, NO, UK, others | No specific iCCS technique defined | Need to expand CCS discussion to heavy industry, iCCS benefits at global level and greater risks at local level, learning from public engagement research literature |

[1] ICQ = Information-Choice Questionnaire; [2] N/A = not available; [3] PESTEL = Political, economic, social, technological, environmental, legal.

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
