# Peer review of "Social Acceptance of Carbon Capture and Storage (CCS) from Industrial Applications"

_sustainability, doi:10.3390/su132112278_

Round 1

Reviewer 1 Report

  • The author present an up-to-date topic
  • the paper is well elaborated and presented
  • There is rigour in methods and background
  • Maybe the author could refer to the Convention on Biological Diversity (https://www.cbd.int/doc/c/326e/cf86/773f944a5e06b75dfc5866bf/sbstta-23-03-en.pdf) and its limitation to iCCS
  • Table 2 is based on the journals format requirements, but it is not really functional in terms of understanding and overview - so, I recommend using names instead of numbers or even better add the authors' names before the numbers
  • In the discussion section, the author should address points, issues, facts missing, e.g rebound effects or systemic risks
  • the limitation section should be widened
  • Before publication, please ask for professional proofreading

Author Response

Dear Reviewer,

Thank you very much for your valuable comments, which certainly help to improve the quality of the article. I have tried to implement your suggestions as best I can in the short time available. Please see the attachment.

With best regards,
Katja Witte

Reviewer 2 Report

The submitted material can be considered as the original work. The absolute similarity index of this work, excluding bibliographies and citations, is 11% and is mainly related to termininoology, which is acceptable. Reading this material, however, I felt a certain deficiency in the presentation of the Author's attitude to the technical, technological and environmental aspects of CCS and iCCS processes and the possible threats resulting from these processes. So it would be beneficial to supplement the material in this regard. Social feelings depend significantly on the society's understanding of the essence of CCS processes and the real threat resulting from these processes. The author used 83 bibliographic items in her review, including the work shown in item 26. The similarity index to this work is 1%. This work includes an analysis of similar issues, hence the author's thesis that the presented work is the first review of the literature in the field of CCS acceptance is not entirely true. The author should therefore clearly define what should be understood by the concept of iCCS in relation to the also defined concept of CCS. I also expect clarification of the conclusions in the assessment of the legitimacy of conducting iCCS processes and showing the type and scope of work increasing the social acceptance of these processes, and if the Author proves the environmental viability of the implementation of these processes. I consider it necessary to introduce these definitions. Taking into account the above comments will increase the substantive value of the presented material.

Author Response

(The authors gave the same response as above.)
